# A Study on the Characteristics of Cu–Mn–Dy Alloy Resistive Thin Films

**Ho-Yun Lee [1], Chi-Wei He [2], Ying-Chieh Lee [2,*] and Da-Chuan Wu [3]**

[1] Department of Material Science and Engineering, École polytechnique fédérale de Lausanne (EPFL), Lausanne 1010, Switzerland; Lillian9620227@gmail.com

[2] Department of Materials Engineering, National Pingtung University of Science &Technology, Pingtung 91201, Taiwan; d7346203@gmail.com,

[3] ZEUS International Management Consultant Company, Kaohsiung 80147, Taiwan; DCWU0614@PCHOME.COM.TW

[*] Correspondence: YCLee@mail.npust.edu.tw; Tel.: +886-8-7703202 (ext.7556)

**Abstract:** Cu–Mn–Dy resistive thin films were prepared on glass and $Al_2O_3$ substrates, which was achieved by co-sputtering the Cu–Mn alloy and dysprosium targets. The effects of the addition of dysprosium on the electrical properties and microstructures of annealed Cu–Mn alloy films were investigated. The composition, microstructural and phase evolution of Cu–Mn–Dy films were characterized using field emission scanning electron microscopy, transmission electron microscopy and X-ray diffraction. All Cu–Mn–Dy films showed an amorphous structure when the annealing temperature was set at 300 °C. After the annealing temperature was increased to 350 °C, the MnO and Cu phases had a significant presence in the Cu–Mn films. However, no MnO phases were observed in Cu–Mn–Dy films at 350 °C. Even Cu–Mn–Dy films annealed at 450 °C showed no MnO phases. This is because Dy addition can suppress MnO formation. Cu–Mn alloy films with 40% dysprosium addition that were annealed at 300 °C exhibited a higher resistivity of ~2100 μΩ·cm with a temperature coefficient of resistance of –85 ppm/°C.

**Keywords:** CuMn alloy; dysprosium; thin film resistors; resistivity; TCR

## 1. Introduction

One of the fundamental passive components is the thin film resistor, which is applied primarily in electronic circuits. Thin film resistors have exceptional properties, such as a low temperature coefficient of resistance, high precision, high stability and low noise, and are commonly used in precision electronic equipment [1–3].

Copper–manganese alloy films with high thermal stability and low resistance are usually used in mobile electronic devices. Copper allows for low electrical resistivity and manganese produces thermal stability in this Cu–Mn alloy system [4]. Based on the equilibrium phase diagram, there are no intermetallic phases in the Cu–Mn system. Copper can serve as a substitute species for Mn in the FCC (Face Centered Cubic) lattice [5]. A characteristic feature of Mn in Cu is its larger activity coefficient compared to other elements that have limited solubility, such as Al and Mg [6]. This feature makes the Cu–Mn solid solution less stable than other Cu alloys when a stable reaction, such as Mn oxidation, can take place. Thus, Mn does not tend to precipitate or segregate within the Cu film but can easily diffuse out to the surface and interface under oxidative conditions [7].

Misjak et al. reported on the specific resistivity of Cu–Mn films, which they measured over the whole composition range [8]. The resistivities of pure Cu and Mn films were 1.7 μΩ·cm and 174 μΩ·cm, respectively. The curve increased monotonically when the Mn content was 0–80 at.%, with a maximum of 205 μΩ·cm at 80 at.% Mn that corresponds to a temperature coefficient of resistance

(TCR) of −308 ppm/°C [8]. Focusing on the electrical properties of $Cu_{0.5}$–$Mn_{0.5}$ alloy films, the resistivity was about 137 μΩ·cm with a TCR of −377 ppm/°C. However, the TCR value was too high, which would have to be improved for applications in mobile electronic devices.

In order to minimize the resistance change in Cu–Mn alloy films, foreign elements are added to improve the TCR. The addition of dysprosium was attempted to improve the electrical properties of Cu–Mn resistive films since rare earth doping has been used as an effective way to regulate the electrical properties of oxides [9-11]. Dysprosium has a larger resistivity (92.6 μΩ·cm) and a higher melting point (1407 °C). Dy may be beneficial for the enhancement of resistivity and the minimization of TCR in thin films. The effects of Dy content and annealing temperature on the phases, microstructural and electrical properties of Cu–Mn thin films are investigated in this study.

## 2. Experimental Procedure

### 2.1. Cu–Mn–Dy Thin Film

Cu–Mn–Dy thin films with a thickness of 80 nm were prepared on substrates using a direct current (DC) and radio frequency (RF) magnetron co-sputtering system. A $Cu_{0.5}$–$Mn_{0.5}$ alloy and dysprosium with a diameter of 76.2 mm were used as targets (purities of 99.95% and 99.9%, respectively). The $Cu_{0.5}$–$Mn_{0.5}$ alloy target was set at the DC position. The dysprosium target was set at the RF position. To obtain different dysprosium contents in the $Cu_{0.5}$–$Mn_{0.5}$ alloy film, the DC power was fixed at 50 W, and the RF power was changed from 40 W to 70 W. A background pressure of $4 \times 10^{-7}$ torr was maintained using a cryo-pump in the sputtering chamber. The sputtering of argon gas with a purity of 99.999% at a flow rate of 60 sccm was executed using mass flow controllers, and the working pressure was maintained at $3 \times 10^{-3}$ torr. In order to measure TCR, Cu–Mn films were deposited onto polished $Al_2O_3$ substrates. These alumina substrates with cell sizes of $1.6 \times 0.8$ mm² were used in this study. Glass with dimensions of $20 \times 10$ mm² and silicon wafers with dimensions of $10 \times 10$ mm² were used for the sheet resistance measurements and thin film thickness, respectively.

### 2.2. Analysis

Alloy films deposited onto the glass substrate at room temperature were subjected to microstructure (transmission electron microscopy) and phase evolution (X-ray diffraction) analysis. Alloy films deposited onto $Al_2O_3$ substrates with Ag electrodes were used to measure the resistance and TCR. The as-deposited films were annealed at 250–450 °C for 2 h at a heating rate of 5 °C/min in a $N_2$ atmosphere. $N_2$ gas with a purity of 99.99% was used in this study.

The sheet resistance $R_s$ is commonly used to characterize materials made by thin film deposition, which was measured using the four-point probe technique. The thickness of Cu–Mn–Dy films after co-sputtering was measured using field-emission scanning electron microscopy (FE-SEM, Hitachi S-4700 Japan, Tokyo, Japan) on a cross-section of the film. The resistivity measurement of specimens using the four-probe method was consistent with the resistivity obtained by the $R_{sheet}$ and film thickness "$t$". The resistance of the specimens was measured using a digital multimeter (HP 34401A, Santa Clara, CA, USA) at different temperatures. The specimens were soaked for 6 minutes at t he set temperatures before measuring the electrical properties while the TCR values of the specimens were calculated using Equation (1) with the resistance at 25 °C and the resistance at 125 °C [1]:

$$TCR = [(\Delta R/\Delta T) \times 1/R] \times 10^6 \; ppm/K \tag{1}$$

X-ray diffraction (XRD, Bruker D8A Germany, Mannheim, Germany) and an electron probe micro-analyzer (EPMA, JEOL JXA-8900R Electron Probe X-ray Microanalyzer, Tokyo, Japan) were used to determine the crystalline phases and the compositions of the thin films, respectively. Microstructural, selected area diffraction (SAD) patterns and EDS (energy dispersive spectroscopy) analysis of the specimens were executed using a field-emission transmission electron microscope

(FE-TEM, FEIE.O. Tecnai F20, Eindhoven, The Netherlands) equipped with an energy dispersive spectrometer at an accelerating voltage of 200 kV.

## 3. Results and Discussion

Cu–Mn–Dy alloy films were deposited onto the substrates using a DC and RF magnetron co-sputtering system. The compositions of the as-deposited film were examined using the electron probe micro-analyzer (EPMA). The relative concentrations of copper, manganese and dysprosium were analyzed at three points in the deposited films. The measured data for copper, manganese and dysprosium are listed in Table 1. The dysprosium content was increased with an increase in the RF sputtering power in the Cu–Mn films. For example, Dy increased from 20.5 at.% at 40 W to 40.0 at.% at 70 W.

**Table 1.** Compositions of Cu–Mn–Dy thin films prepared at a direct current (DC) 50 W with different radio frequency (RF) powers sputtered on cooper sheet.

| Power (W) | DC/RF | 50/40 | 50/55 | 50/70 |
|---|---|---|---|---|
| | Cu | 52.3 | 45.5 | 37 |
| Element (at.%) | Mn | 27.2 | 24.5 | 23 |
| | Dy | 20.5 | 30 | 40 |

Figure 1 shows X-ray diffraction patterns of the as-deposited and annealed Cu–Mn films with different added amounts of dysprosium, which were deposited on glass substrates. Annealing was performed in a N$_2$ atmosphere at a temperature up to 450 °C. All of the Cu–Mn–Dy films annealed at ≤350 °C displayed an amorphous structure, except for alloy films without Dy addition (Figures 1a,b). When the annealing temperature was set at 450 °C, a Cu crystallization phase with a (111) peak was clearly observed in the Cu–Mn–Dy films. An additional (200) diffraction peak of the Cu$_2$O phase appeared in the Cu–Mn–Dy films after annealing at 450 °C as shown in Figure 1c. This demonstrates that none of the elements were crystallized or oxidized in Cu–Mn films with Dy addition after annealing at 300 °C in a N$_2$ atmosphere. However, two crystallization phases (Cu and MnO) appeared in Cu–Mn alloy films without Dy addition as shown in Figure 1. The crystallization strength of the MnO peaks also increased with increased annealing temperatures. This is believed to be due to the oxidation of the Cu–Mn alloy film during annealing. Iijima et al. reported that Mn does not tend to precipitate or segregate within the Cu film but can easily diffuse out to the surface and interface under oxidative conditions [6,12]. In contrast, Mn at the interface can be selectively oxidized by reacting with oxygen under proper conditions. In our study, the samples were annealed in a N$_2$ atmosphere. Why did the MnO cause oxidation in the alloy films? Further studies of the microstructure of alloy film with transmission electron microscopy will be performed and discussed later.

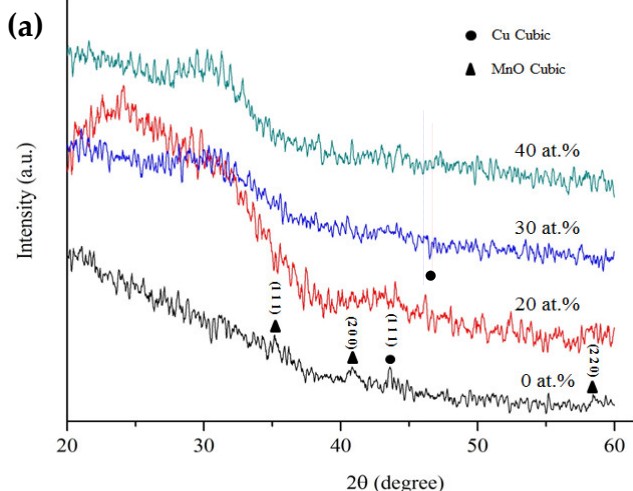

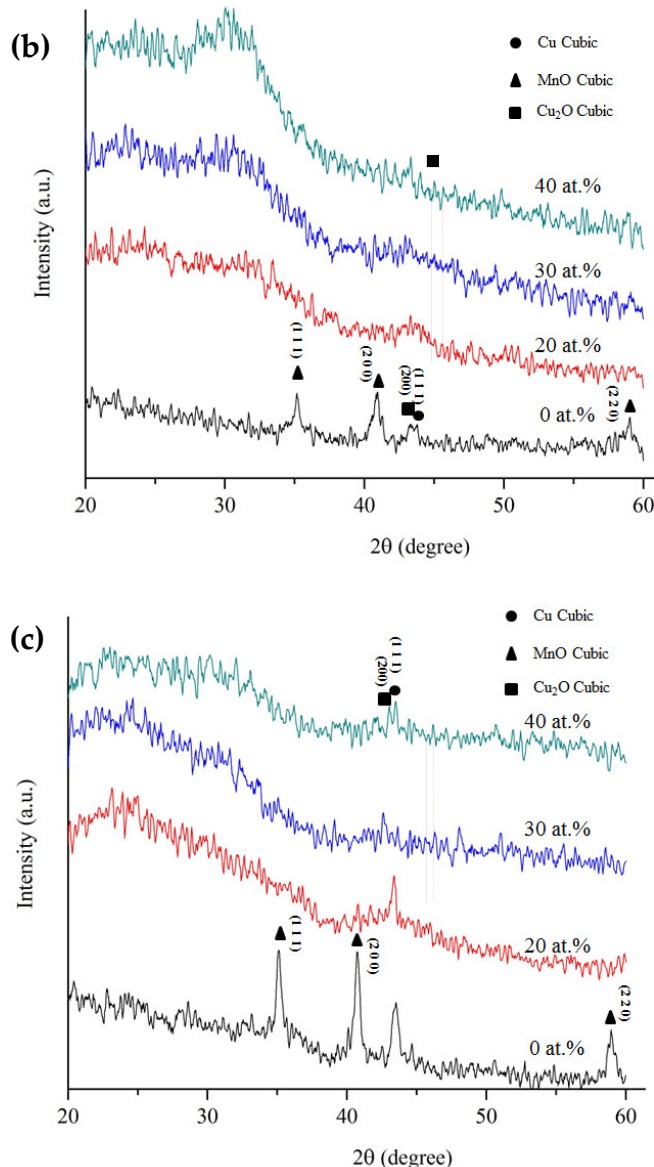

**Figure 1.** X-ray diffraction patterns of Cu–Mn–Dy thin films with various amounts of dysprosium addition annealed at (**a**) 300 °C; (**b**) 350 °C; and (**c**) 450 °C.

Figure 2 shows a cross-section TEM micrograph, SAD and Energy Dispersive X-ray Analysis EDX analysis of Cu–Mn films without Dy addition that was annealed at 300 °C. The film thickness is about 60 nm as shown in Figure 2a. This shows that there are different crystalline structures between the upper and lower layers in Cu–Mn alloy films. In the lower layer, MnO crystallites are present as demonstrated using SAD analysis (Figure 2c). Regarding MnO formation, it is believed that oxygen from the $Al_2O_3$ substrate reacted with Mn during annealing because the MnO phase exists in the lower layer of the alloy film. In the upper layer, the $Cu_2O$ phase was found according to SAD analysis as shown in Figure 2d. This result indicates that the Cu–Mn alloy film surface can be oxidized to cause the formation of the $Cu_2O$ phase during annealing in a $N_2$ atmosphere. Under $N_2$ annealing, the surface of Cu–Mn alloy films was oxidized due to the presence of oxygen and/or moisture inside the chamber/furnace [13,14]. Copper only forms two thermodynamically stable oxides, namely CuO and $Cu_2O$. From the Gibbs free energy point of view, the $Cu_2O$ phase likely formed first because the $Cu_2O$ phase ($\Delta G_0 = -122$ KJ/mol) formation has a lower Gibbs free energy than the CuO phase ($\Delta G_0 = -99$ KJ/mol) [15,16]. Moreover, Luo et al. reported that only $Cu_2O$ is expected to form at very low oxygen partial pressures [17]. A similar oxidation phenomenon

occurred for Cu–Mn alloy films annealed in pure Ar at 350 °C for 1800 s. On the other hand, the $Cu^{2+}$ ion has a greater charge density than the $Cu^+$ ion; thus, it forms much stronger bonds that release more energy. $Cu^{2+}$ is more stable than $Cu^+$ in an aqueous medium. Wang et al. reported that $Cu^{2+}$ cations react with metallic Cu to form $Cu^+$ through a disproportionation reaction, with these unstable $Cu^+$ cations subsequently rapidly reacting with $O–O_{(adsorb)}$ to form $Cu_2O$ [18].

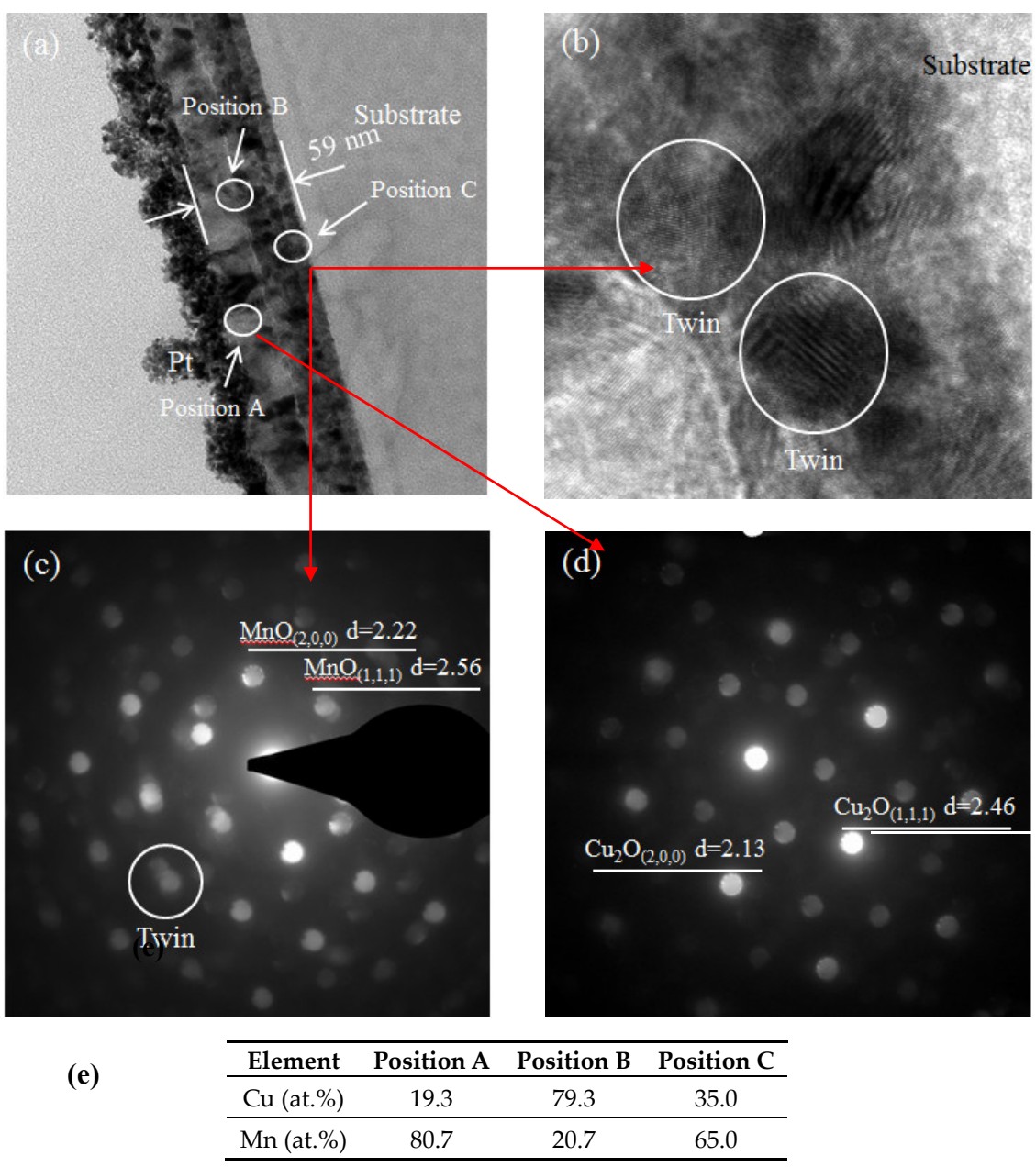

| Element | Position A | Position B | Position C |
|---|---|---|---|
| Cu (at.%) | 19.3 | 79.3 | 35.0 |
| Mn (at.%) | 80.7 | 20.7 | 65.0 |

**Figure 2.** TEM micrographs of Cu–Mn films annealed at 300 °C: (**a**) bright field of cross section; (**b**) HRTEM micrographs; (**c**) selected-area electron diffraction at position C; (**d**) selected-area electron diffraction at position A; (**e**) Energy Dispersive X-ray Analysis EDX analysis.

The sample surface shows the formation of a discontinuous oxide layer [6]. The EDX analysis conducted using different positions in the Cu–Mn film is listed in Figure 2e. The Cu/Mn ratios differed greatly between the bottom, middle and surface areas of the film as they were 0.24, 3.4 and 0.54, respectively. This result indicates that the copper atoms are concentrated in the middle of the film while the manganese atoms are distributed on the surface and bottom. Haruhiko Asanuma et al. [12] reported that Mn migrates toward the interface and reacts with a surface oxide layer until, finally, a Mn complex oxide layer is formed during subsequent annealing [19,20].

Figure 3 displays a cross-section TEM micrograph, EDX and SAD analysis of Cu–Mn alloy films with 40 at.% Dy addition that was annealed at 300 °C. There is an oxidation layer with a thickness of 3 nm on the surface as shown in Figure 3a. The oxidation layer can also be called a passivation layer. However, an amorphous structure was observed in the alloy films, which is evident from the halo SAD patterns and high-resolution transmission electron microscopy as shown in Figures 3b,c. This result is consistent with the XRD analysis (Figure 1). It was noted that the Cu–Mn–Dy thin films had an amorphous structure at 300 °C. Unlike the Cu–Mn alloy films, there were some crystalline phases observed after annealing at 300 °C This may be attributed to the multiple element alloy effect, which could be explicated by the kinetics theory because of slow atomic diffusion [21,22]. This means that dysprosium addition in Cu–Mn films leads to the formation of an amorphous structure and oxidation resistance after annealing at 300 °C. The EDX analyzed positions in the Cu–Mn–Dy film are listed in Figure 3d. This table shows that the element distribution for Cu, Mn and Dy was more uniform between the top and bottom parts of the film.

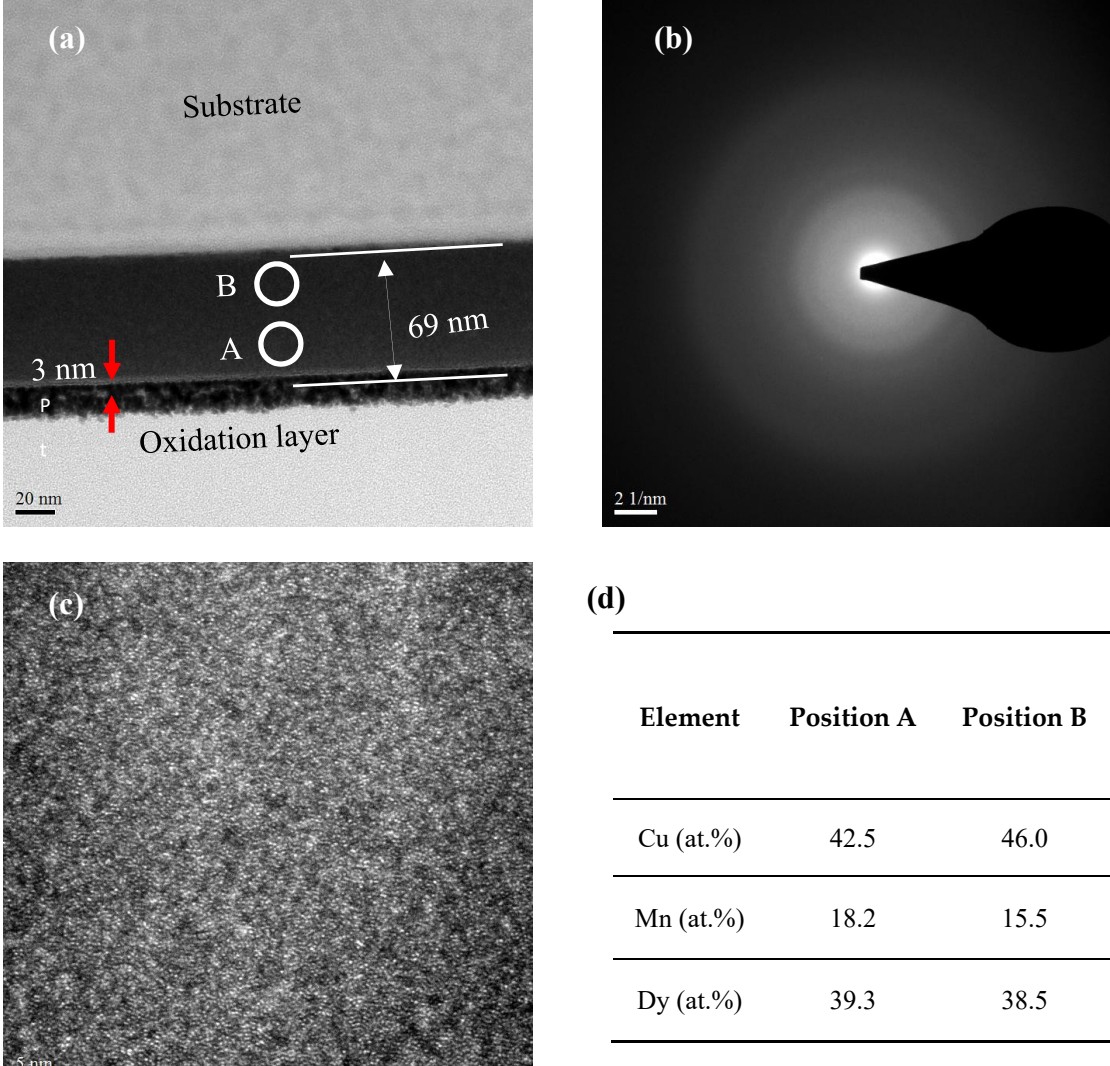

| Element | Position A | Position B |
|---|---|---|
| Cu (at.%) | 42.5 | 46.0 |
| Mn (at.%) | 18.2 | 15.5 |
| Dy (at.%) | 39.3 | 38.5 |

**Figure 3.** TEM micrographs of Cu–Mn films with 40 at.% Dy addition annealed at 300 °C: (**a**) bright field of cross section; (**b**) selected-area electron diffraction; (**c**) HRTEM micrographs; (**d**) EDX analysis.

Figure 4 shows a cross-section TEM micrograph, EDX and SAD analysis of Cu–Mn films with 40 at.% Dy addition that was annealed at 350 °C. When the annealing temperature was increased to 350 °C, there was an oxidation layer with a thickness of 5 nm on the surface as shown in Figure 4a. A 5-nm crystalline layer on the film bottom was observed using high-resolution transmission electron

microscopy analysis as shown in Figure 4b. According to the SAD pattern analysis, this crystalline layer belongs to the Cu crystallization structure, as shown in Figure 4c. It is known that amorphous thin films are in a metastable state. The heat treatments can initiate ordering processes, such as structural relaxation and crystallization [23]. To further observe and determine the size of precipitated particles, HRTEM was used. Some microcrystallites appeared on the surface films, as shown in Figure 4d. The precipitated particles were distributed on the amorphous matrix and had a size smaller than 5 nm. The nanobeam electron diffraction pattern is shown in Figure 4e, which demonstrates that these microcrystallites belong to the $Cu_2O$ phase. The EDX was analyzed using different positions in the Cu–Mn–Dy film, as shown in Figure 4f. Unlike Cu–Mn alloy films (Mn atoms are distributed on the surface and bottom), we found that Mn atoms had a more uniform distribution in the Cu–Mn–Dy annealed films compared to Cu–Mn annealed films.

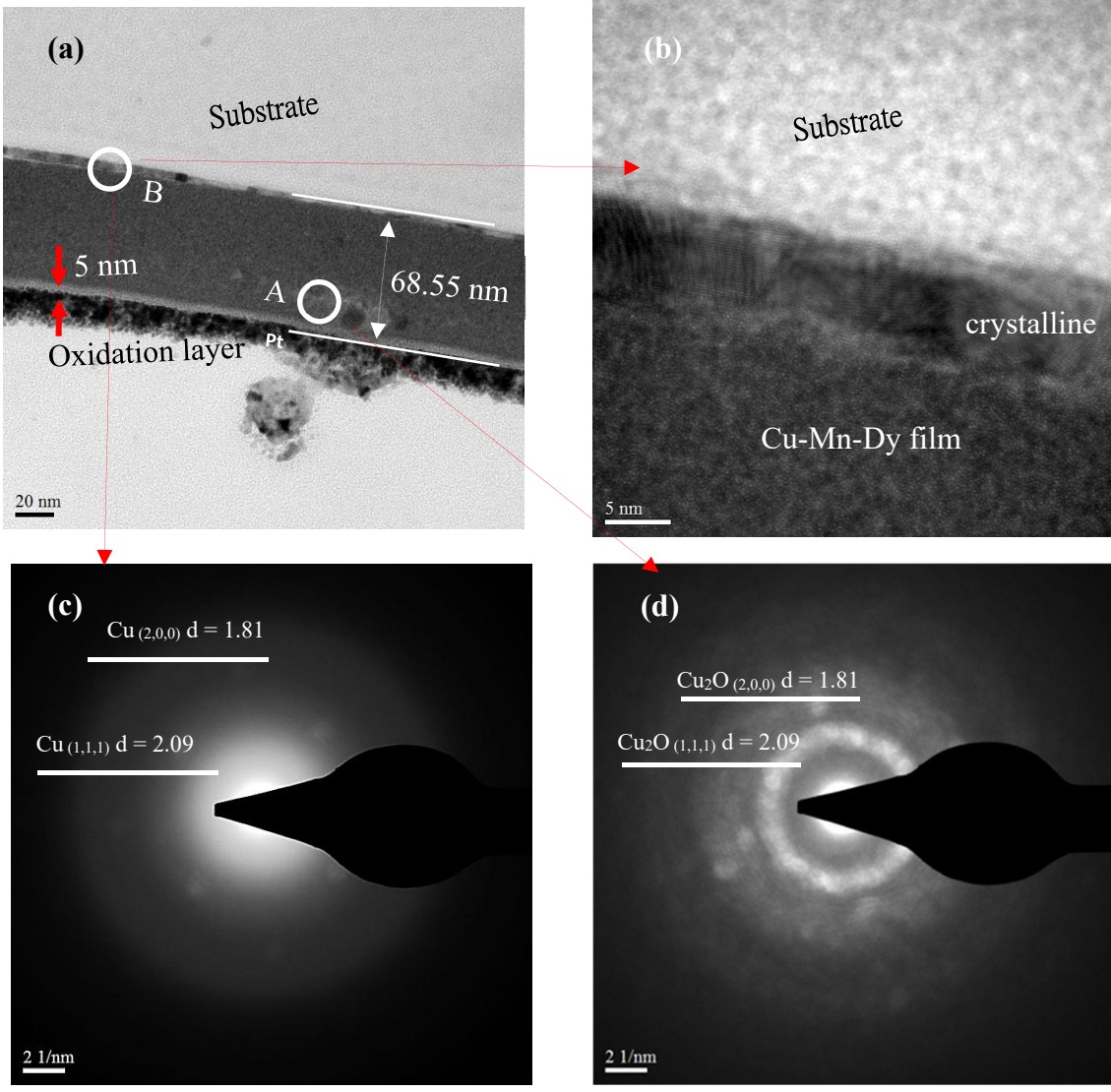

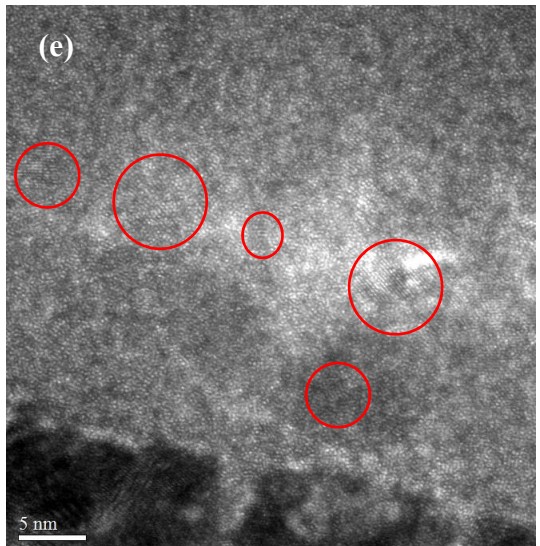

**Figure 4.** TEM micrographs of Cu–Mn films with 40 at.% Dy addition annealed at 350 °C: (**a**) bright field of cross-section, (**b**) HRTEM micrographs at position B, (**c**) selected-area electron diffraction of position B, (**d**) selected-area electron diffraction of position A, (**e**) HRTEM micrographs at position A, (**f**) EDX analysis.

| Element | Position A | Position B |
|---------|-----------|-----------|
| Cu (at.%) | 40.0 | 69.9 |
| Mn (at.%) | 17.6 | 13.9 |
| Dy (at.%) | 42.4 | 16.2 |

The X-ray photoelectron spectroscopy (XPS) analysis included survey scans to understand the elemental composition at different depths. The compositions of the Cu–Mn–Dy film with 40 at.% Dy addition that was annealed at 300 °C were analyzed using electron spectroscopy for chemical analysis (ESCA) as listed in Table 2. It was noted that the oxygen concentration significantly differed between the film surface and interior area (at a depth of 28 nm). The oxygen concentrations for the film surface and interior area were 68.9 at.% and 1.6 at.%, respectively. This result indicates that the surface of Cu–Mn–Dy film was oxidized. The same result was obtained by TEM analysis as shown in Figure 3a.

**Table 2.** Compositions of Cu–Mn films with 40 at.% Dy addition annealed at 300 °C using electron spectroscopy for chemical analysis (ESCA).

| Film position / Element | Surface | Internal |
|---------|---------|----------|
| Cu (at.%) | 8.8 | 47.6 |
| Mn (at.%) | 7.4 | 9.3 |
| Dy (at.%) | 14.9 | 41.5 |
| O (at.%) | 68.9 | 1.6 |

Figure 5 shows the effects of dysprosium addition on the electrical properties of annealed Cu–Mn films. The resistivity of Cu–Mn films increased with an increase in dysprosium. Dy addition enhanced the resistivity of Cu–Mn films. There are two reasons to explain this phenomenon: (1) dysprosium has a higher resistivity (92.6 μΩ·cm) that contributes to the alloy film resistivity and (2) an amorphous structure in the alloy film can be obtained by Dy addition. However, it was noted that the resistivity of Cu–Mn–Dy film increased significantly after annealing at 350 °C. This is due to the increase in the oxidation of the alloy film after annealing at 350 °C compared to 300 °C since $Cu_2O$ microcrystallites exist at 350 °C (Figure 4e). The resistivities of Cu–Mn films with 40 at.% Dy addition were ~2100 and ~1200 μΩ·cm after annealing at 250 °C and 300 °C, respectively. Generally speaking, the resistivity of alloy film increases with an increase in the annealing temperature because the grain boundaries, crystal defects and oxides generation are increased after annealing [3]. Nevertheless, many scattering behaviors are believed to affect the resistivity of Cu–Mn–Dy films, which includes surface scattering, grain boundaries scattering, rough surfaces scattering, impurities

scattering and oxides generation [24]. However, we hypothesize that the generation of oxides plays an important role in determining the electrical properties of Cu–Mn–Dy films.

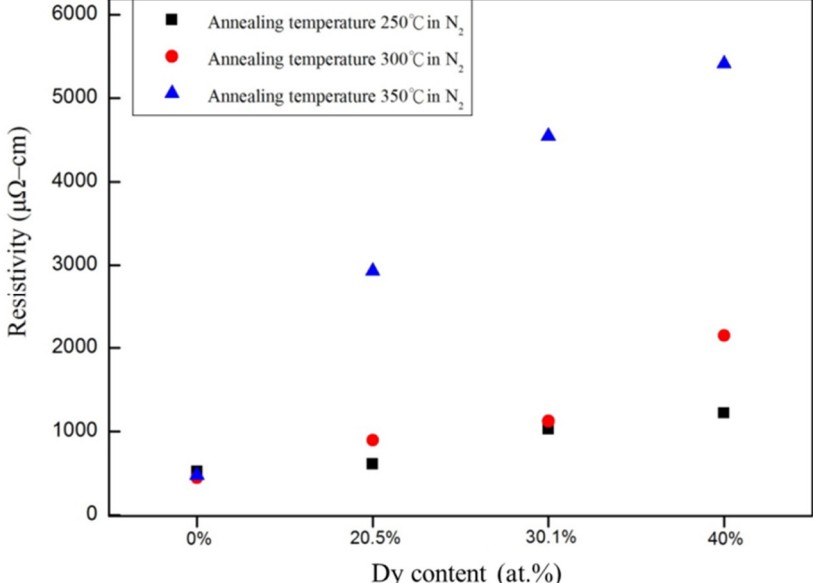

**Figure 5.** Room temperature resistivity of Cu–Mn films with various Dy contents annealed at different temperatures.

Figure 6 shows the effects of dysprosium addition and annealing temperature on the temperature coefficient of resistivity (TCR) of Cu–Mn films in a nitrogen atmosphere at ambient temperatures for 2 h. The annealing treatment is an important factor in determining the resistor stability of the thin film. When the annealing temperature is ≤300 °C, the TCR changes significantly from approaching zero to having a positive value with increasing Dy content, except for specimens with 40 at.% Dy addition. As the annealing temperature was increased to 350 °C, the TCR values rapidly increased, except for specimens with 40 at.% Dy addition, which may be caused by the presence of Cu microcrystallites. As the alloy films crystallized, the TCR became positive because the majority of all metals have a positive TCR [25,26]. However, the TCR values suddenly changed from positive to negative when the Dy addition was increased to 40 at.%. This may be due to the increased $Cu_2O$ formation on the surface (producing an oxidation layer), as shown in Figures 1 and 4. This result implies that the TCR value is strongly dependent on Dy content in the Cu–Mn films. The Cu–Mn films with 40 at.% Dy addition displayed a resistivity of ~2100 $\mu\Omega\cdot$cm with the smallest TCR (−85 ppm/°C) after annealing at 300 °C in a $N_2$ atmosphere.

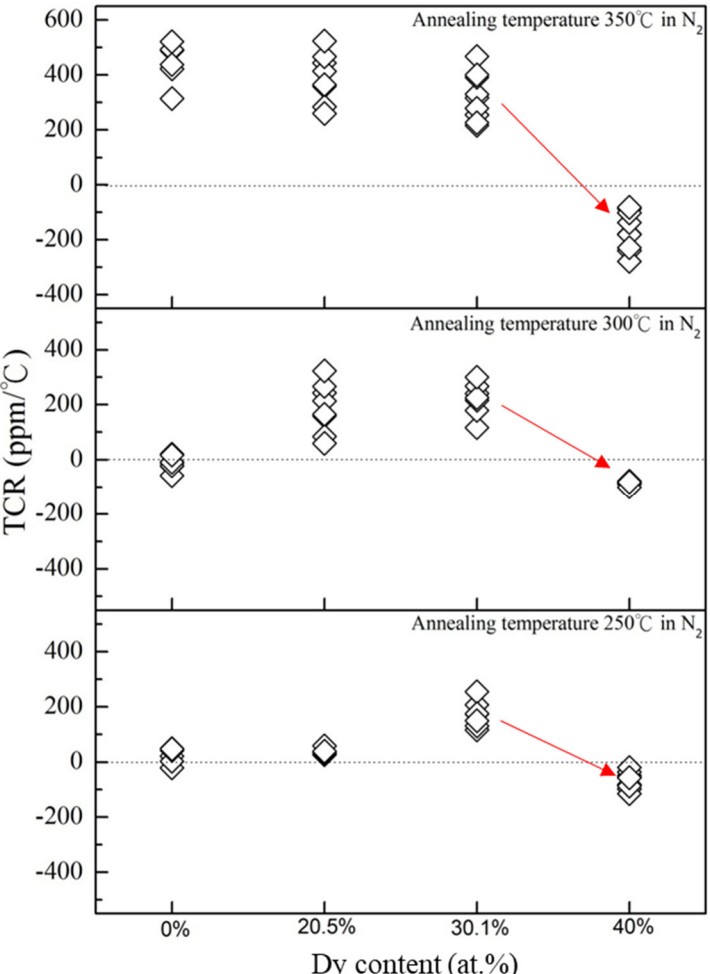

**Figure 6.** Temperature dependence of the temperature coefficient of resistance (TCR) of Cu–Mn films with various Dy contents annealed at different temperatures.

## 4. Conclusion

In this study, we prepared Cu–Mn–Dy thin films for the purpose of fabricating thin film resistors with high resistivity and low TCR. The effects of annealing temperature and Dy content on the electrical properties of the Cu–Mn–Dy thin film were investigated. Our conclusions are summarized as follows.

For the Cu–Mn resistive thin films, Cu and MnO crystalline phases existed when the annealing temperature was 300 °C. When the annealing temperature was ≥350 °C, Cu, MnO and $Cu_2O$ crystalline phases were presented in the Cu–Mn films.

For the Cu–Mn–Dy resistive thin films, all Cu–Mn–Dy films annealed at ≤300 °C had an amorphous structure. This means that Dy addition in Cu–Mn alloy films can suppress the Cu crystalline formation and Mn oxidation. However, there was still the Cu crystalline phase on the bottom part of the film and $Cu_2O$ phase on the film surface when the annealing temperature was 350 °C. The Cu–Mn films with 40 at.% Dy addition that were annealed at 300 °C exhibited a resistivity of ~2100 μΩ·cm with the smallest temperature coefficient of resistance (−85 ppm/°C).

**Author Contributions:** Data curation, H.-Y.L.; Formal analysis, C.-W.H.; Funding acquisition, D.-C.W.; Methodology, Y.-C.L.

**Funding:** This research received no external funding.

**Acknowledgments:** The authors would like to acknowledge the financial support of this research from the Ministry of Science and Technology of Taiwan under contract No. 106-2221-E-020-009.

**Conflicts of Interest:** The authors declare no conflict of interest.

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
