# Peer review of "A Study on the Characteristics of Cu–Mn–Dy Alloy Resistive Thin Films"

_coatings, doi:10.3390/coatings9020118_

Reviewer 1 Report

This is an interesting article but some important discussion is missing.

Defect structures and also the oxidation states of metal ions are missing. For instance, Cu is in which form Cu+ or Cu2+. Why authors do not investigate or mention this. And point defects or other kinds of defects are not mentioned properly.

Some revision is necessary.

Author Response

- Defect structures and also the oxidation states of metal ions are missing. For instance, Cu is in which form Cu+ or Cu2+. Why authors do not investigate or mention this. And point defects or other kinds of defects are not mentioned properly.

Ans: Thanks for comments.

On the other hand, the Cu2+ ion has a greater charge density than the Cu+ ion and so forms much stronger bonds releasing more energy. Cu2+ is more stable than Cu+ in aqueous medium. Wang et al. reported that Cu2+ cations react with metallic Cu to form Cu+ through a disproportionation reaction, these Cu+ cations are unstable and will rapidly react with the O–O(adsorb) to form Cu2O [19].

[19] Zhifeng Wang, Chunling Qin, Li Liu, Lijuan Wang, Jian Ding, Weimin Zhao, Synthesis of CuxO(x=1,2)/Amorphous Compounds by Dealloying and Spontaneous Oxidation Method, Materials Research 17(1):33-37 · February 2014

Reviewer 2 Report

This paper presents a detailed study on the Cu-Mn-Dy resistive thin film characteristics. The authors have doped the Cu-Mn with Dy at studied there structural, morphological, crystalline, and resistive properties at different annealing temperatures. After thorough revision of the manuscript, I found many errors that need to be addressed before publication. Hence, I want to reconsider the manuscript after major revision of the comments provided in the next. I believe the comments, if addressed properly, would improve the overall quality of the manuscript for the wide readers of Coatings.

Major Comments:

1)      There are many structural and grammatical errors in the manuscript. The manuscript needs to be checked with a native English speaker or a Linguistic expert.

2)      The manuscript is more focused to the materials aspect of the compounds. What are the pragmatic applications of the study and how would the study be exploited in the realm of application oriented research and development on the material?

3)      Line 44-45: How come the electrical properties be improved when the resistivity is increased by the addition of Dy? Also, it’s better to support the claim by providing the I-V characteristics of the films for better understanding and flow.

4)      Generally, FESEM is used to understand the clear and concise morphology of nanostructures. Did the authors performed FESEM analysis of the films? If yes then I prefer adding the results into the manuscript for clarity. Otherwise, express the reason why authors avoid using FESEM for morphology analysis?

5)      Figure 2(a): Certain morphological structure are visible at the background beyond 59 nm film. What are those structures and how would it affect the overall performance of the films?

6)      The description for Figure 2 is quite vague and incomprehensible. It’s better to explain the data one by one for different parts of the Figure and then make a correlation between all the Figure parts. Also, express all the information revealed and highlighted in the Figure for readers' understanding.

Minor Comments:

1)      The abbreviations such as FESEM, XRD, TEM, and TCR have not been used again in the abstract hence need to be deleted.

2)      Use MDPI Coatings' formal template to cite references into the manuscript.

3)      For readers' convenience, provide a bibliographic reference to Eq. 1.

4)      Define all the abbreviations at the first instance of their usage in the main body of the manuscript.

5)      Why Figure 2(d) has a double panel (d)?

Author Response

- Major Comments:

1) There are many structural and grammatical errors in the manuscript. The manuscript needs to be checked with a native English speaker or a Linguistic expert.

Ans: Thanks for comments. Done!

I have sent to MDPI for English editing.

2) The manuscript is more focused to the materials aspect of the compounds. What are the pragmatic applications of the study and how would the study be exploited in the realm of application oriented research and development on the material?

Ans: Thanks for comments

The thin film resistor is one of the fundamental components used primarily in electronic circuits. The demand for thin film resistors with low temperature coefficients of resistance (TCR) and high precision has dramatically increased in recent years.

An important technical parameter of thin film resistors is the temperature coefficient of resistance (TCR). A high TCR will result in the resistance value drifting, influencing the resistor accuracy as the temperature changes.[6] The main factors influencing TCR include the film composition, sputtering process and annealing temperature. The film composition plays a decisive role among these three factors. Therefore, employing an appropriate method for depositing a suitable film composition is essential to obtaining high-resistance resistors with a low TCR.

The construction of thin film resistor is shown as in Fig. a1, we make (5) Thin film resistive material. To measure the resistivity and TCR, Cu-Mn films were deposited onto polished Al2O3 substrates. These alumina substrates with 1.6´0.8 mm cell sizes were used in this study. The Al2O3 substrates had Ag electrodes printed on the surface.

We can make the resistive films by sputtering, and thin film resistors can be prepared by passive components company as shown in Fig. a2.

Fig. a1: The construction of thin film resistor.

Fig. a2: The thin film resistor.

3) Line 44-45: How come the electrical properties be improved when the resistivity is increased by the addition of Dy? Also, it’s better to support the claim by providing the I-V characteristics of the films for better understanding and flow.

Ans: Thanks for comments

The dysprosium has larger resistivity (92.6 mΩ-cm) and higher melting point (1407 °C). Dy may be beneficial for enhancement of resistivity and minimization of TCR in thin films.

Also, it’s better to support the claim by providing the I-V characteristics of the films for better understanding and flow

Ans: To evaluate the electrical properties of resistive films, the resistivity and TCR are measured usually, rather than I-V characteristic curves. I-V characteristic curves are generally used as a tool to determine and understand the basic parameters of a component or device and which can also be used to mathematically model its behaviour within an electronic circuit. Generally speaking, the resistivity and TCR are represented of resistive film quality whether is able to apply in the thin film resistors.

4) Generally, FESEM is used to understand the clear and concise morphology of nanostructures. Did the authors performed FESEM analysis of the films? If yes then I prefer adding the results into the manuscript for clarity. Otherwise, express the reason why authors avoid using FESEM for morphology analysis?

Ans: Thanks for comments

We use FESEM to analyze the thickness of CuMn alloy films, but it is not easy to observe the difference on the film microstructures as shown in Fig. a3. We use TEM to analyze the microstructure, crystal structure, and composition of CuMn alloy films.

 Fig. a3: FESEM to analyze the thickness of CuMn alloy films

5) Figure 2(a): Certain morphological structure are visible at the background beyond 59 nm film. What are those structures and how would it affect the overall performance of the films?

Ans: Thanks for comments

Fig. 2(a) is revised to easy understand.

Generally speaking, the resistivity of CuMn alloy film increases by increasing the annealing temperature, because the oxides generation was increased after at 300 °C annealing [23].

Fig. 2(a): bright field of cross section

6) The description for Figure 2 is quite vague and incomprehensible. It’s better to explain the data one by one for different parts of the Figure and then make a correlation between all the Figure parts. Also, express all the information revealed and highlighted in the Figure for readers' understanding.

Ans: Thanks for comments

Figure 2 is a cross-section TEM micrograph, SAD and EDX analysis of Cu-Mn films without Dy addition annealed at 300 °C. The film thickness is about 60 nm as shown in Fig. 2(a). It can be seen there are different crystalline structures between upper and lower layer in the CuMn alloy films. In lower layer part, MnO crystallites are evidenced using SAD analysis as shown in 2(c). Regarding MnO formation, it is believed that oxygen came from the Al2O3 substrate to react with Mn during annealing because the MnO phase exists in the alloy film bottom layer. In upper layer part, the Cu2O phase was existed according to SAD analysis, as shown in Fig. 2(d). This result indicates that the Cu-Mn alloy film surface can be oxidized to cause Cu2O phase formation during annealing in N2 atmosphere. Under N2 annealing, the surface of CuMn alloy films were oxidized due to the presence of some oxygen content and/or moisture inside the chamber/furnace [14,15].

Minor Comments:

1) The abbreviations such as FESEM, XRD, TEM, and TCR have not been used again in the abstract hence need to be deleted.

Ans: Thanks for comments. Done!

2) Use MDPI Coatings' formal template to cite references into the manuscript.

Ans: Thanks for comments. Done!

3) For readers' convenience, provide a bibliographic reference to Eq. 1.

Ans: Thanks for comments. Done!

4) Define all the abbreviations at the first instance of their usage in the main body of the manuscript.

Ans: Thanks for comments. Done!

5) Why Figure 2(d) has a double panel (d)?

Ans: Thanks for comments. Done!

Reviewer 3 Report

The manuscript by H.-Y. Lee et al. presents fabrication and characterization of Cu-Mn-Dy resistive thin films. Through systematic experiments, the authors show that the addition of Dy in Cu-Mn films increases their amorphous nature when annealed at < 300C. The authors demonstrate Cu-Mn films with 40 wt. % Dy displayed the resistivity ∼2100 micro-ohm-cm with smallest TCR -85 ppm/C after annealing at 300C in N2 atmosphere. The results presented in the manuscript are interesting and the manuscript is worth publication in the Coatings journal if the following concerns of the reviewer are addressed: 

1.       The authors explain the relevance of their work in the introduction section. However, can the authors explain how their work is a significant advance in the field? For example, how does resistivity and TCR obtained by the authors here compare to the previously reported values?

2.       Line 172, The oxygen concentrations for film surface and interior are 1.6 at. % and 68.9 at.%, respectively. The oxygen concentration numbers for film surface and interior are interchanged and should be corrected.

3.       Figures 2, 3, 4, the authors should not combine tables with figures. Similar to Table 1, these tables should also be included separately.  

4.       The authors may consider doing optical characterization of the thin films. It would be interesting to see if the films would be optically thin enough to allow transmission of light.

5.       Line 13, it should be field.

6.       Line 39, it should be monotonically.

7.       Line 170, what is ECSA?

8.       The manuscript requires moderate English editing at multiple places before final publication.

Author Response

Reviewer 3

1. The authors explain the relevance of their work in the introduction section. However, can the authors explain how their work is a significant advance in the field? For example, how does resistivity and TCR obtained by the authors here compare to the previously reported values?

Ans: Thanks for comments. Done!

In this study, the Cu-Mn films with 40 at.% Dy addition annealed at 300 °C exhibited a resistivity of ~2100 mW-cm with smallest temperature coefficient of resistance (-85 ppm/°C). Based on the electrical properties, the resistivity of CuMnDy alloy films is higher than NiCrMnZr or NiCrMnY alloy films. In our previous study is shown as below:

NiCrMnZr films exhibited the smallest temperature coefficient of resistance (+53 ppm/°C) with the resistivity 510 ?Ω-cm after annealing at 300 °C in air.[a1] NiCrMnY films exhibited a resistivity ~480 mW-cm with the temperature coefficient of resistance (TCR) at +30 ppm/°C. However, NiCrMnYNb films exhibited the smallest temperature coefficient of resistance (+5 ppm/°C) with the resistivity 585 ?Ω-cm after annealing at 300 C in air.[a2]

a1: Cheng-Hsien Lin, Ho-Yun Lee, Yaw-Teng Tseng, Ying-Chieh Lee, A study on the NiCrMnZr thin film resistors prepared using the magnetron sputtering technique, Thin Solid Films, 2018, 660, 695.

a2. Wei-Ju Chen, Tung-Yueh Liu, Ho-Yun Lee, Ying-Chieh Lee, Ni-Cr-Mn-Y-Nb resistive thin film prepared by co-sputtering, Materials Chemistry and Physics, 2018, 210, 327.

2. Line 172, The oxygen concentrations for film surface and interior are 1.6 at. % and 68.9 at.%, respectively. The oxygen concentration numbers for film surface and interior are interchanged and should be corrected.

Ans: Thanks for comments. Change it!

3. Figures 2, 3, 4, the authors should not combine tables with figures. Similar to Table 1, these tables should also be included separately. 

Ans: Thanks for comments. Change it!

4. The authors may consider doing optical characterization of the thin films. It would be interesting to see if the films would be optically thin enough to allow transmission of light.

Ans: Thanks for comments. This is good suggestion.

However, I am sorry, we don’t have the equipment for measuring optical characterization. We can evaluate it in the future experiments.

5. Line 13, it should be field.

Ans: Thanks for comments. Change it!

6. Line 39, it should be monotonically.

Ans: Thanks for comments. Change it!

7. Line 170, what is ECSA?

Ans: Thanks for comments. Done!

8. The manuscript requires moderate English editing at multiple places before final publication..

Ans: Thanks for comments.

I have sent to MDPI for English editing.

Round  2

Reviewer 1 Report

The manuscript can be now accepted.

Reviewer 2 Report

The overall quality of the manuscript has been improved after the revisions. I believe the manuscript would be beneficial for the wide readers of Coatings. Hence, I want to accept the manuscript in present form.